# Comprehensive Evaluation of Water Resource Characteristics in the Northern Yangtze River Delta, China

**Liang He** [1,2,*,†] , **Chenfang Xu** [1,†] , **Shaohua Lei** [3,*] , **Ling Chen** [1] and **Suozhong Chen** [2]

1  School of Environmental Science, Nanjing Xiaozhuang University, Nanjing 211171, China
2  Key Laboratory of Virtual Geographic Environment, Nanjing Normal University, Ministry of Education, Nanjing 210023, China
3  State Key Laboratory of Hydrology-Water Resources and Hydraulic Engineering, Nanjing Hydraulic Research Institute, Nanjing 210029, China
*  Correspondence: heliang@njxzc.edu.cn (L.H.); shaohualei@nhri.cn (S.L.)
†  These authors contributed equally to this work.

**Abstract:** The Yangtze River Delta is one of the most economically developed regions on the eastern coast of China. However, a local imbalance currently exists between its water resource environment and economic and social development due to its rapid urbanization. Thus, the reasonable evaluation and protection of local water resources are necessary. This study explores the northern Yangtze Delta, which is a more developed water system, as a pilot area. The temporal and spatial variation characteristics of rainfall and evaporation and their influencing factors were analyzed on the basis of 29 surface water sampling points, 16 rainfall stations, and three evaporation stations in the field from 1956 to 2019. Accordingly, the overall water supply quality of the river basin, the availability of different water resources, and the application of evaluation methods were assessed. Results show that local precipitation and evaporation are characterized by uneven spatial and temporal distributions in local areas, which, in turn, leads to the uneven temporal distribution of runoff, increasing the imbalance between the availability and demand of the limited local water resources. Nevertheless, the overall performance of local water quality is good. Surface water quality is mostly II to III, and locally IV. Most noncompliant months are during the non-flood season, and all values exceed the standard permanganate index. Groundwater is Class III or better, and the hydrochemistry type is predominantly calcium bicarbonate, sodium bicarbonate, and magnesium bicarbonate. By exploring the evaluation model of the Yangtze River Delta watershed characteristics, this study aims to provide a helpful reference for extending water resource evaluation in the Yangtze River Delta. Accordingly, this study can promote the sustainable development of the economic and social sectors of the Yangtze River Delta and the construction of its ecological environment.

**Keywords:** groundwater; surface water; water resource evaluation; water quality analysis; northern Yangtze River Delta

## 1. Introduction

Water is a fundamental resource for human survival, an economic resource for social development, and an essential ecological resource for supporting the cyclical operation of ecological systems [1]. As the birthplace of the "two mountains" theory (i.e., clear water and green mountains are gold and silver mountains), the Yangtze River Delta region is currently one of the most urbanized, economically developed, and densely populated regions in China. In addition, the Yangtze River Delta is the source of the eastern route of China's South-to-North Water Transfer Project [2,3]. However, the rapid development of society has posed a serious challenge to the local water environment [4,5]. Therefore, encouraging the sustainable use of water resources and ensuring the coordinated development of the basin require a comprehensive evaluation of the Yangtze River Delta's water resource features [6,7].

Groundwater resource assessment is an important part of water resource assessment [8]. At present, many methods for evaluating groundwater resources have been proposed locally and overseas [9,10]. From the perspective of water source investigation and evaluation, these methods can be divided into two categories: practical analysis and mathematical analysis methods [11]. Practical analysis methods calculate groundwater resources through various tests and by using relatively simple mathematical means. Examples include the hydrogeological comparison method based on the evaluation of similar principles, the water balance method, and the extraction test method based on the water balance method [12,13]. Meanwhile, mathematical analysis methods use mathematical theories to calculate groundwater resources. Examples include probabilistic statistical analysis, groundwater analysis, and systematic analysis, which are based on actual observations; and hydrodynamic analysis, numerical methods, and electrical network simulation, which are based on seepage theory and field investigation tests [14–16]. However, the aforementioned traditional methods for obtaining groundwater resource data suffer from problems, such as high acquisition costs, low calculation efficiency, large evaluation errors, and subjectivity [17]. Suitable evaluation methods should be selected in accordance with the specific hydrogeological conditions and the detail level of the given information to achieve better results. By assessing the amount of groundwater resources, a basis can be provided for the protection and rational allocation of these resources, along with important technical support for the rational and efficient development of groundwater resources [18].

In addition, groundwater resource assessment is a systematic and complex project that not only assesses the amount of groundwater contained in an area or basin, but also evaluates and analyzes water quality [19–21]. The selection of a suitable water quality assessment method is an essential part of the water quality assessment process. A reasonable water quality assessment should be able to provide water quality categories, major pollution factors, and spatial and temporal changes in water quality [22,23]. The single-factor evaluation method, pollution index method, fuzzy evaluation method, gray system evaluation method, analytic hierarchy process, artificial neural network method, and water quality identification index method are common examples [24,25]. In particular, the single-factor evaluation method is the most widely used technique. It assesses the lowest of the single-factor water quality classes as the result leading to its conclusion of overprotection; it is applied to the study of safeguarding the water ecological environment. The pollution index method can visually determine whether the combined water quality meets the functional area objectives; however, it cannot identify the combined water quality category [26]. When the comprehensive water quality level is I to V, the fuzzy evaluation method, gray system evaluation method, analytic hierarchy process, artificial neural network method, and water quality identification index method provide the same evaluation conclusion [27–29]. However, when the integrated water quality level is V (poor), the evaluation conclusions of the fuzzy evaluation method, gray system evaluation method, analytic hierarchy process, and artificial neural network method are conservative [30]. Only the water quality identification index method solves the continuous description problem of poor water quality, enabling a scientific and reasonable evaluation of this type of water [31]. In summary, many techniques are used to assess water quality, and the benefits and drawbacks of their results vary; therefore, selecting a variety of techniques for a comprehensive examination is typically more logical [32].

The current study thoroughly assesses and analyzes the water resource features of the Yangtze River Delta pilot area by using data from numerous sources, including underlying surface, rainfall, and evaporation, along with field sampling data from surface and groundwater bodies. In this manner, the latest water quantity and quality trends in the basin are obtained, and a water resource assessment model applicable to the Yangtze River Delta basin is explored [33–35]. The findings can be used to offer technical guidance to water resource management and development in the Yangtze River Delta basin. Meanwhile, water resource protection measures can be developed by local management authorities to achieve a harmonious coexistence between humans and nature in the Yangtze River Delta.

## 2. Materials and Methods

### 2.1. Study Area

The pilot area is located in Gopo Lake, northern Yangtze River Delta, between 119°08′24″ E–119°24′36″ E and 33°01′12″ N–33°18′36″ N. The total area of the region is 267.92 km$^2$, of which the water area is 123.85 km$^2$, accounting for 46.23% of the total area. The terrain is high in the west and low in the east, belonging to the plain area, with a ground elevation of about 4.00–5.00 m. The climate belongs to the northern subtropical subhumid monsoon zone, with four distinct seasons. Winter is dry and cold; summer is hot and rainy; and spring and autumn are dry, wet, cold, and warm.

The pilot area belongs to the Huai River basin, and its water system belongs to the Grand Canal system in the lower reaches of the Huai River basin, with interlocking rivers and lakes in the area and a network of water. The north–south spreading rivers are largely the Beijing–Hangzhou Grand Canal and the Central Pai River. The east–west flowing rivers are mostly the North Yunxi River, Shanyang River, Dazhai River, Zhonggang River, Main Pai River, and South Yunxi River. The Beijing–Hangzhou Grand Canal is located on the east side of the pilot area, running north to south, and is a river basin with a length of 35.53 km in the pilot evaluation area and an average water level of 6.50 m. The western side of the pilot evaluation area is dotted with Baima Lake and Baoying Lake, which are connected to the Yangtze River via the Grand Canal. Together, these rivers and lakes form the main water system network of the pilot evaluation area, with multiple functions, such as flood control, drainage, and irrigation. Through field investigation, diving water quality sampling points were laid at an average of 10 km$^2$ in the pilot area, and 23 diving water quality sampling points and 6 surface water quality sampling points were selected. The map of the pilot area and its geographical location are shown in Figure 1.

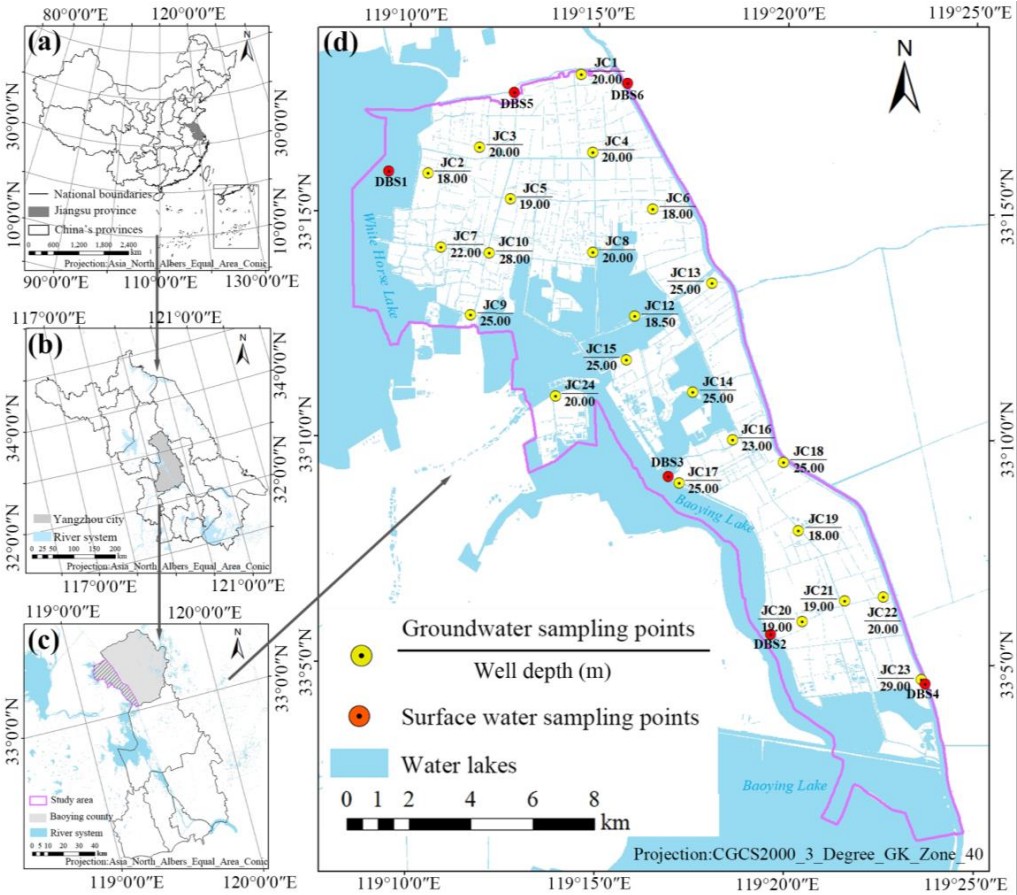

**Figure 1.** Map of the pilot area and its geographical location. (**a**) Jiangsu Province, China. (**b**) Yangzhou City. (**c**) Baoying County. (**d**) Spatial location of water quality sampling points in the pilot area.

*2.2. Data Sources*

2.2.1. Underlying Surface and Groundwater Level

Data on the underlying surface of the pilot area were provided by the Natural Resources Bureau of Baoying County, Yangzhou City, Jiangsu Province, China. To reflect the consistent characteristics of the runoff series, the underlying surface was divided into individual types in accordance with flow production characteristics, and then flow production models were developed separately. In addition, surface water resources were calculated daily. The various types of land areas in the production flow model were adopted from the latest survey statistics of local land administration authorities. The results basically reflect the condition of the recent (horizontal year 2019) underlying surface of the pilot area, as indicated in Table 1.

**Table 1.** Pilot area's statistical table of underlying surface conditions (2019).

| Region | Area (km²) | Statistics | City Town Village | | Water | | Paddy Field | Dryland |
| | | | Hardening | Non-Hardening | Water Surface | Other | | |
|---|---|---|---|---|---|---|---|---|
| Huaihe River (Gopo Lake) | 267.92 | Area (km²) (%) | 14.73 5.50 | 9.50 3.54 | 83.87 31.31 | 39.98 14.92 | 95.22 35.54 | 24.62 9.19 |

The amount of water in transit was calculated on the basis of the established national hydrological stations as control sections. Groundwater information was obtained from the natural resource department for monitoring groundwater wells, monitoring information from the Yangzhou Branch of the Jiangsu Provincial Water Resource Survey Bureau, and data from the current water quality survey.

2.2.2. Rainfall and Evaporation

When studying the characteristics of rainfall in the region, 16 rainfall stations in the pilot area were selected for the analysis by considering the length of the data series and the principle of uniform distribution of rainfall stations, with a time series of 1956 to 2019. For rainfall stations with less than 64 years of data series, synchronous data from rainfall stations adjacent to them with at least 64 years of data series were used to establish a correlation, interpolate, and extend to 64 years. Furthermore, the observed data from 1980 to 2019 (40 years) from three evaporation stations in Liuzha, Xinghua, and Huangqiao were used to analyze the evaporation capacity of the pilot area. However, the three evaporation stations have evaporators other than $E_{601}$ and $\varnothing 80$. Considering that the evaporation measured by the $E_{601}$ evaporator better reflects evaporation capacity and facilitates analysis and comparison, the observed values of different evaporator models were converted and unified into the $E_{601}$ model in the current assessment. The rainfall and evaporation stations are spatially distributed as shown in Figure 2.

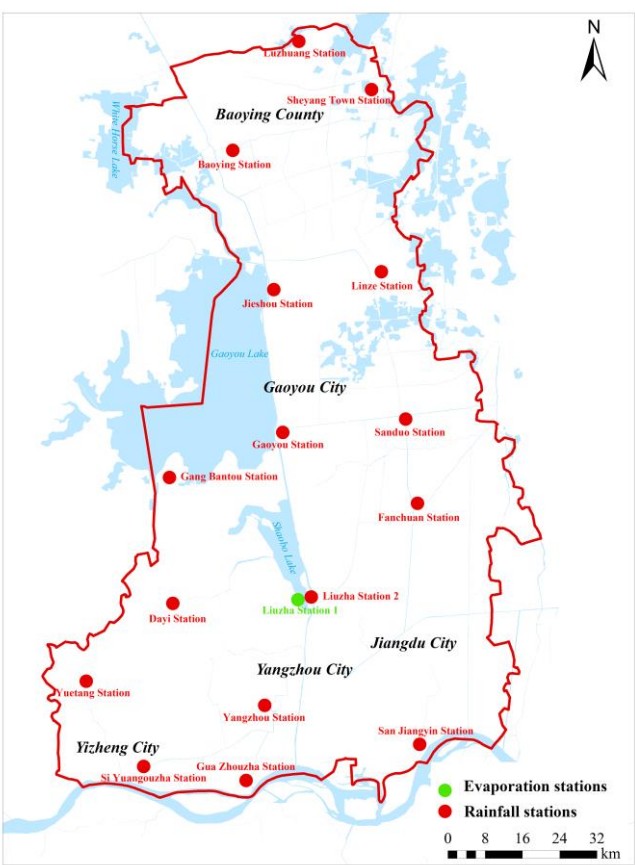

**Figure 2.** Spatial distribution of rainfall and evaporation stations.

*2.3. Methods*

2.3.1. Synchronous Series Representative Correlation Methods

A stochastic series of meteorological data generally has one or more complete cycles of abundance and depletion. The relative stability of the series depends crucially on the relative stability of the statistical parameters, the coefficient of variation (Cv), and the bias coefficient ($C_S$) [36]. A representative analysis of the series is required to reduce the error due to the potential for sampling error in the Cv and statistical parameters over the course of a long series. By using the comparative analysis method of long and short series statistical parameters, one to three long series ($n > 60$) observation stations in the city (county) area were selected for comparative analysis to determine the stability of the proposed series statistical parameters. The rainfall modal ratio coefficient cumulative difference product curve method was used. First, a good representative long series of observation station data was selected, and then the mean value of the series ($\overline{P}$) was determined. Finally, the rainfall modal ratio coefficient was calculated separately for each year. The formula is as follows:

$$Ki = Pi/\overline{P}, \tag{1}$$

where $\overline{P}$ is the mean value of the rainfall series, $Pi$ is the annual rainfall for each year, and n is the length of the series for each year. Then, to calculate the cumulative yearly value $C$ from the beginning to the end of the data, the formula is as follows:

$$C = \sum_{n}^{i=1}(Ki - 1). \tag{2}$$

Lastly, the *C–t* (year) cumulative curve was plotted. Rising cycles in the cumulative difference curve are periods of abundance, while falling cycles are periods of depletion. By using the cumulative annual average method, each year's average was determined and then

analyzed. When the average no longer exhibits a large trend of changes is representative. The calculation formulas are as follows:

$$\overline{P}1 = P1, \tag{3}$$

$$\overline{P}2 = \frac{P1 + P2}{2}, \tag{4}$$

$$\overline{P}3 = \frac{P1 + P2 + P3}{3}, \tag{5}$$

where $P1$, $P2$, $P3$, $\cdots$, $Pn$ are the annual rainfall amounts in order of precession from the status quo year; while $\overline{P}1$, $\overline{P}2$, $\overline{P}3$, $\cdots$, $\overline{P}n$ are the cumulative annual averages of rainfall. The analysis can generally be broken down into periodic and stochastic characterization, with periodic characterization generally for longer series.

### 2.3.2. Water Interpolation Calculation Methods

When calculating the diversion volume along a river, if the sluice gates have hydrological stations, then the measured water level and flow rate are used to determine the relevant line and deduce the diversion volume of each gate. Interpolation calculations are performed when encountering years of missing information or locks without information [37–39].

For medium-sized gates with a long series of data, the correlation analysis method is used for interpolating the diversion and drainage volume. For example, when the characteristics and functions of the sectional gates are similar, the missing stations are extrapolated from the information of actual stations.

For small and medium-sized gates without data, the interpolation of the diversion capacity is performed using the unit net width diversion capacity extrapolation method. By utilizing data on the characteristics of the gates, the limited number of measurements in certain years of these gates is compared with the simultaneous measured data of the controlled medium-sized gates. The conversion factor (c) between the uninformed small and medium-sized gates and the controlled medium-sized gates is determined to derive the drainage capacity of the uninformed small and medium-sized gates.

### 2.3.3. Water Availability Estimation Method

Surface water resource availability estimates can be divided into reverse calculation and positive methods [40]. Southern regions with more abundant water resources and coastal solitary rivers into the sea are generally used for positive methods. For mountainous areas where the development and utilization of water resources in the upper reaches of large rivers or tributaries are more difficult, the utilization of water resources is largely restricted by the construction of water supply projects. Meanwhile, water supply capacity is also limited to a certain extent. The formula for calculating the number of water resources available is as follows:

$$W_{\text{surface water availability}} = k_{\text{water consumption factor}} \times W_{\text{maximum water supply}} \cdot \tag{6}$$

For the lower reaches of large rivers, the major factor that determines the extent of their water use is the magnitude of demand, and the corresponding formula is as follows:

$$W_{\text{surface water availability}} = k_{\text{water consumption factor}} \times W_{\text{maximum water demand}} \tag{7}$$

### 2.3.4. Water Quality Analysis Method

Within the pilot area, the surface water pH of the Grand Canal, Yunxi River, Baoying Lake, and Baima Lake was tested, along with the groundwater pH of 23 groundwater wells. The assay was commissioned to the Analysis and Testing Center of Nanjing Normal University. The assay was performed with an inductively coupled plasma atomic emission spectrometer (00304214), an ultraviolet–visible absorption spectrometer (20060475), and an ion chromatograph ThermoICS-900 (00185109). By using the single-factor evaluation

method, the water quality categories of the surface water and groundwater were determined by implementing the criteria of the Surface Water Environmental Quality Standard (GB3838-2002) for evaluation. In addition, several evaluation indicators were selected for $Na^+$, $K^+$, $Ca^{2+}$, $Mg^{2+}$, $Cu^{2+}$, Fe, Pb, Cd, Cr, As, fluoride, chloride, sulfate, nitrite, nitrate, ammonia N, hexavalent Cr, total P, bicarbonate, carbonate, mineralization, and total hardness. Then, Alyokin and Shukarev classifications were combined to determine the water chemistry types of the surface water and groundwater.

## 3. Results and Analysis

### 3.1. Rainfall Characteristic Analysis

As shown in Figure 3, the multiyear average annual rainfall contour maps for 1956–2019 and 1980–2019 were derived from line spacing contours of 20 mm and combined with multiyear rainfall data for the pilot area. The range of rainfall contours in the Yangtze River Delta region from 1956 to 2019 was between 960 mm and 1020 mm, with the range of annual average rainfall contours less than 980 mm and a decreasing trend of annual average rainfall from east to west. The range of rainfall contours from 1980 to 2019 was 960–1040 mm. The range of contours was still less than 980 mm, with a higher average annual rainfall in the east than in the west, and a decreasing trend in average annual rainfall from east to west. In addition, the shape and position of the 1000 mm contour changed in a completely different manner, but the shape and position of the 1020 mm contour remained largely unchanged.

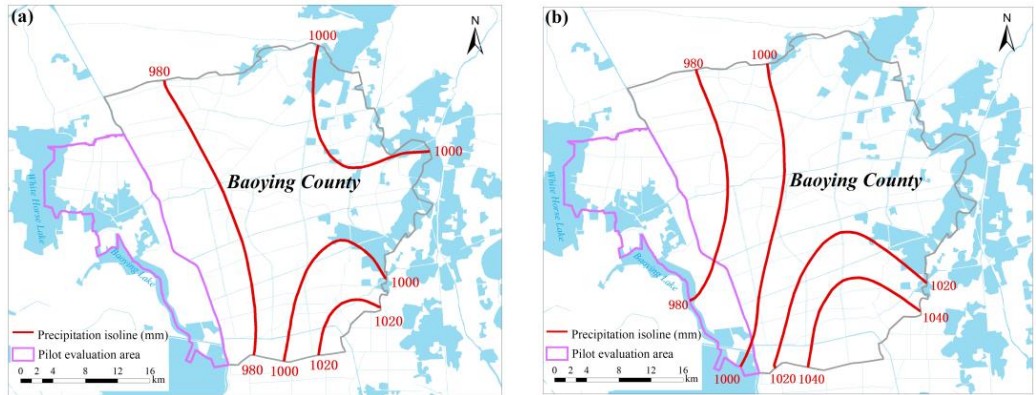

**Figure 3.** Map of Precipitation Contour. (**a**) 1956–2019. (**b**) 1980–2019.

Using Baoying Station as the representative rainfall station in the pilot area, statistical analysis was performed on its rainfall data by month over the years. Presenting the series from 1956 to 2019 as an example, the maximum four consecutive months of rainfall occurred from June to September, with 63.31% of the annual rainfall. Although the non-flood period was seven months, the rainfall at that time accounted for less than 36.69% of the annual rainfall.

As shown in Figure 4, the data from 16 rainfall stations in the pilot area were selected and combined using the simultaneous series representative correlation method to produce a map of the cumulative mean and difference distribution of rainfall in the pilot area.

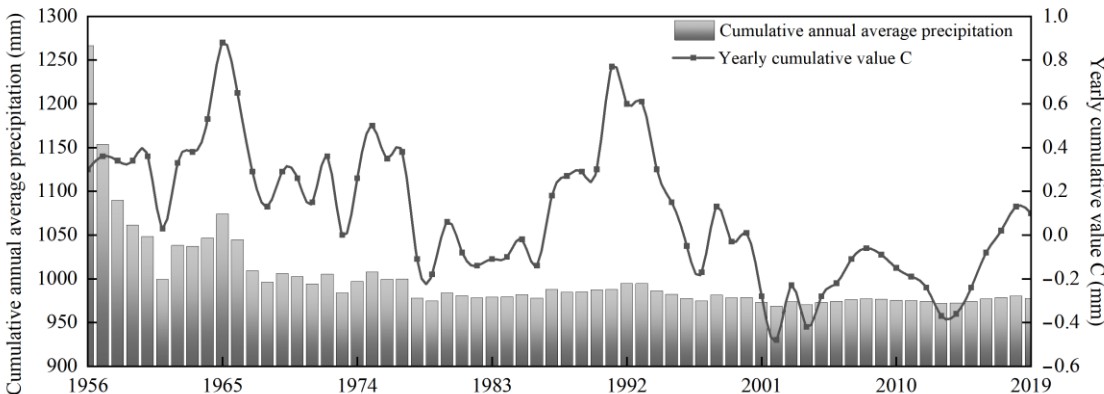

**Figure 4.** Cumulative annual average precipitation and yearly cumulative value C.

*3.2. Evaporation Characteristic Analysis*

The multiyear average monthly maximum water surface evaporation in the pilot evaluation area accounts for 12.9% of the annual evaporation, and the monthly minimum evaporation accounts for 3.1% of the annual evaporation. In addition, the maximum four consecutive months of evaporation totaled 432.0 mm, accounting for 47.9% of the annual evaporation.

As shown in Figure 5, given that evaporation in the pilot area was less before 1980 compared with after 1980, the Six Gate station was used as a representative to analyze the simulated process map of evaporation capacity in the pilot area by utilizing trend simulation. In general, evaporation factors are closely related to meteorological elements, such as temperature, rainfall, humidity, sunshine, and wind speed. However, the regional and interannual variations of evaporation are relatively small compared with the rainfall factor. Therefore, the station network density of evaporation stations is considerably smaller than that of rainfall stations. To analyze evaporation capacity under recent subsurface conditions, 40 years of evaporation data from 1980 to 2019 were used.

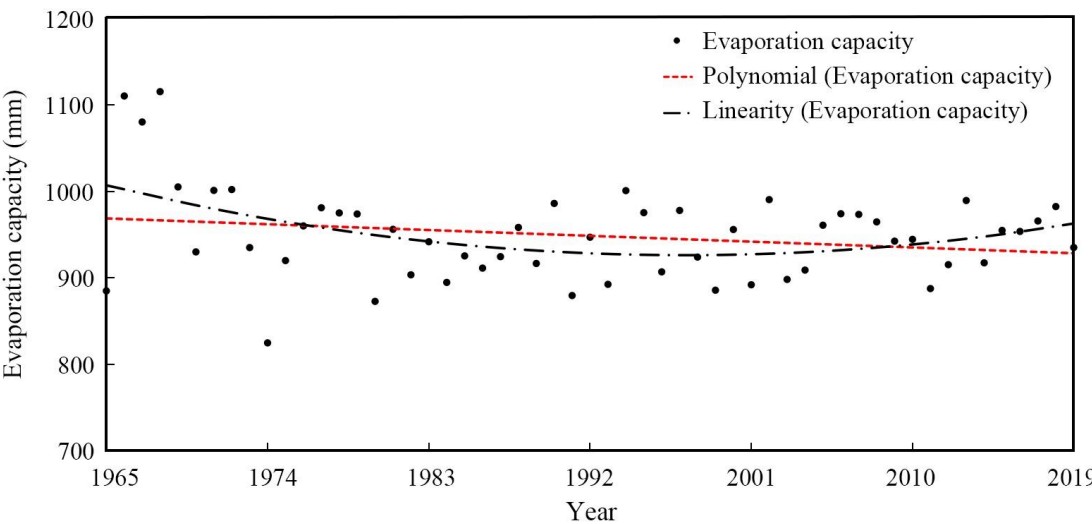

**Figure 5.** Annual evaporation simulation.

*3.3. Available Water Analysis*

3.3.1. Available Surface Water

Surface water resources in the pilot area are mostly formed by local rainfall. The multiyear average annual rainfall for the region is 977.40 mm, with a total rainfall of 262 million m$^3$, a surface runoff depth of 244.0 mm, and a water resource volume of 64 million m$^3$. Calculated using positive and negative algorithms, surface water resources

available in the pilot evaluation area are 14 million m³, with a usable rate of 21.4%. The natural runoff volume from 1956 to 2019 is shown in Figure 6.

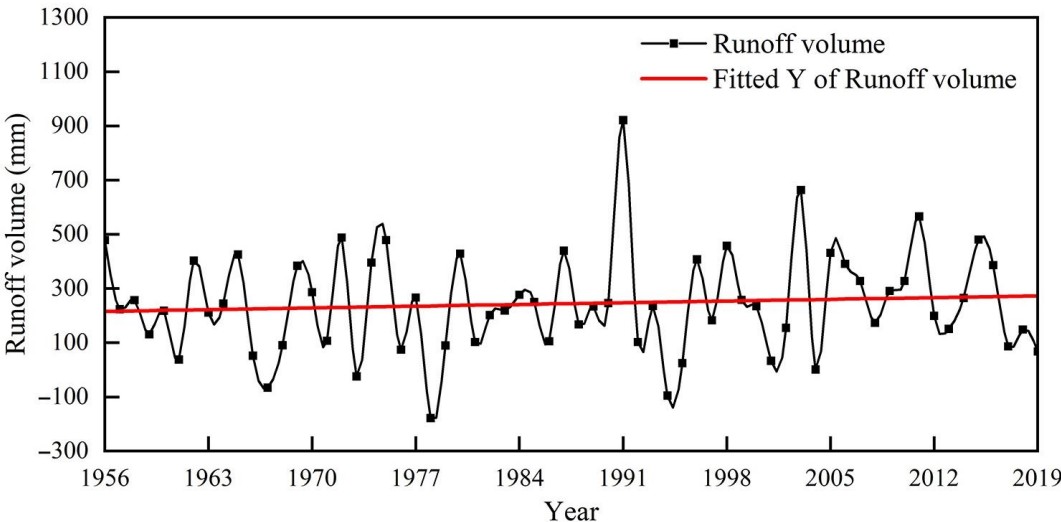

**Figure 6.** Changes in natural runoff from 1956 to 2019.

### 3.3.2. Available Groundwater

The total amount of groundwater resources is the sum of various recharge amounts entering the pilot evaluation area. In particular, atmospheric rainfall naturally recharges about 58.16% of groundwater volume, with the concentration of rainfall during the flood season accounting for about 67% of the annual rainfall. Therefore, this situation results in groundwater recharge during the flood season accounting for 36.33% of the annual recharge. The pilot area is located on a plain, and no baseflow recharge occurs from rivers in the hilly areas in groundwater resource quantity. Thus, no duplicate amount is deducted when determining the amount of groundwater resources. The pilot region has 32.75 million m³ of groundwater resources, and 58.16% of atmospheric rainfall naturally infiltrates into those resources. The amount of groundwater available in the pilot evaluation region is 0.026 billion m³ in accordance with the coefficient approach used to determine the extractable amount of groundwater resources.

### 3.3.3. Available Transit Water

The amount of water in transit is divided into two parts in accordance with the type of water source: Yangtze River and Huai River. The amount of river water that can be used is estimated by calculating the actual use of river water. In the pilot area, the actual available quantity of Yangtze River water includes the amount of water abstracted directly from the Yangtze River for industrial, agricultural production, and domestic use and the amount of river water available in the abdomen area of the Lixia River. On the basis of the actual volume of water abstracted from the Yangtze River, the amount of water available for use by the Yangtze River is estimated by cutting off the amount of water replenished to the river under the guarantee of the downstream water consumption target. By contrast, the water level control method is used to calculate the available quantity of Huai River water. The multiyear average available water resources are determined by calculating the difference between the volume under the monthly multiyear average water level of Gaoyou Lake and Shaobo Lake and the volume under the normal water storage level. The total available volume of water in transit in the pilot area is calculated to be 356 million m³. Of which, 210 million m³ is available for the Yangtze River and 146 million m³ for the Huai River.

### 3.3.4. Total Available Water

The calculation of total water resource availability includes surface water availability, groundwater availability, and transit water availability. In particular, the sum of surface

water availability and groundwater availability minus the amount of reuse provides the total amount of local water resources available for use. In accordance with the calculation result, the total amount of water resources available in the pilot area is 359 million m$^3$.

*3.4. Water Quality Analysis*

Within the pilot area, pH assay results yielded pH values of 7.18 for the Grand Canal (DBS6, DBS4), 7.40 for Baoying Lake (DBS3, DBS2), 6.91 for Baima Lake (DBS1), and 7.26 for the Yunxi River (DBS5). Furthermore, the total hardness of the Grand Canal is 141.25–149.00 mg/L, (soft water). The total hardness of Baoying Lake is 90.25–101.25 mg/L, (soft water). The total hardness of Yunxi River is 161.00 mg/L, (soft water). The total hardness of Baima Lake is 158.25 mg/L, (soft water). One of the surface water information tables is provided in Table 2, with the Yue Tang Reservoir having the smallest content of all ions, followed by the Yangtze River and the Grand Canal, whereas Gaoyou Lake has the largest. Moreover, the mineralization of the Grand Canal, Baoying Lake, the Yunxi River, and Baima Lake was 332.46–341.36, 270.92–365.42, 331.34, and 342.37 mg/L, respectively. This study shows that surface water in the pilot area largely exhibits low mineralization. The water chemistry type of the surface water is mostly calcium bicarbonate, sodium bicarbonate, and magnesium bicarbonate. The water body predominantly contains bicarbonate with a certain amount of Na and K ions. It exhibits low mineralization and total hardness.

**Table 2.** Surface water quality analysis results (Unit: mg/L).

| No. | Depth (m) | pH | Na$^+$ | K$^+$ | Ca$^{2+}$ | Mg$^{2+}$ | Cl$^-$ | HCO$_3$$^-$ | Mineralization | Total Hardness |
|---|---|---|---|---|---|---|---|---|---|---|
| DBS1 | 8.94 | 6.91 | 43.2 | 7.74 | 63.2 | 21.6 | 59.4 | 233 | 342.37 | 158.25 |
| DBS2 | 9.18 | 7.61 | 38 | 4.48 | 36.1 | 20.2 | 54 | 353.8 | 365.42 | 90.25 |
| DBS3 | 8.79 | 7.40 | 40.4 | 4.57 | 40.5 | 21.2 | 53.5 | 153.1 | 270.92 | 101.25 |
| DBS4 | 9.05 | 7.18 | 48 | 7.32 | 56.5 | 18 | 56.6 | 189.7 | 332.46 | 141.25 |
| DBS5 | 9.25 | 7.26 | 45 | 7.4 | 64.4 | 19.5 | 54.1 | 194 | 331.34 | 161 |
| DBS6 | 8.77 | 8.34 | 51 | 8.48 | 59.6 | 19.5 | 56.4 | 192.2 | 341.36 | 149 |

Most of the monitored 23 groundwater wells have pH values between 7.00 and 7.50, which is water quality Classes I–III. Point JC11 is not informative because the data and assays are unavailable given that they were cleared by external factors after the point was set. Groundwater information is provided in Table 3. The total hardness monitoring values for groundwater range from 106.75 mg/L to 342.00 mg/L, with an average value of 216.20 mg/L. The lowest value for total hardness is 106.75 mg/L, which occurs at point JC21, while the highest value is 342.00 mg/L, which occurs at point JC13. Moreover, mineralization monitoring values range from 208.67 mg/L to 1003.94 mg/L, with an average value of 517.81 mg/L. The lowest mineralization value of 308.67 mg/L occurs at point JC5, while the highest value of 1003.94 mg/L occurs at point JC20. The distributions of total groundwater hardness and mineralization contours in the pilot evaluation area are shown in Figure 7. Combined with the single-factor evaluation method, groundwater chemistry types in the pilot evaluation area are predominantly calcium bicarbonate, sodium bicarbonate, and magnesium bicarbonate.

**Table 3.** Groundwater quality analysis results (Unit: mg/L).

| No. | Depth (m) | pH | Na$^+$ | K$^+$ | Ca$^{2+}$ | Mg$^{2+}$ | Cl$^-$ | HCO$_3^-$ | Mineralization | Total Hardness |
|---|---|---|---|---|---|---|---|---|---|---|
| JC1 | 10.26 | 7 | 98.1 | 2.35 | 73.3 | 52.2 | 120.6 | 366.6 | 538.92 | 183.25 |
| JC2 | 8.59 | 7.2 | 52.5 | 2.77 | 103 | 44.2 | 63.6 | 345.3 | 513.44 | 257.5 |
| JC3 | 9.07 | 7.5 | 44.6 | 1.94 | 65 | 27.2 | 66 | 581.9 | 503.68 | 162.5 |
| JC4 | 10.24 | 7.1 | 47.2 | 3.03 | 85.9 | 38.2 | 90.5 | 359.9 | 453.07 | 214.75 |
| JC5 | 10.21 | 7.4 | 39.4 | 1.51 | 51.7 | 23.6 | 13.8 | 351.4 | 308.67 | 129.25 |
| JC6 | 10.41 | 7.1 | 68.5 | 2.44 | 97.7 | 34.7 | 83 | 418.5 | 559.66 | 244.25 |
| JC7 | 9.96 | 7.2 | 48.2 | 2.84 | 102 | 42 | 95.8 | 472.1 | 563.96 | 255 |
| JC8 | 11.68 | 7.1 | 35.6 | 2 | 60 | 32.2 | 11.7 | 428.8 | 356.41 | 150 |
| JC9 | 10.05 | 7.1 | 46.4 | 2.63 | 97.9 | 42.1 | 101.4 | 461.2 | 558.23 | 244.75 |
| JC10 | 10.55 | 7.2 | 47.3 | 1.77 | 63.8 | 42 | 102.5 | 461.2 | 526.25 | 159.5 |
| JC11 | / | / | / | / | / | / | / | / | / | / |
| JC12 | 9.27 | 7.3 | 65.8 | 2.4 | 86.7 | 26.6 | 50.4 | 411.8 | 441.43 | 216.75 |
| JC13 | 10.19 | 7.1 | 61.7 | 3.76 | 137 | 38.3 | 99.7 | 493.5 | 594.16 | 34,200 |
| JC14 | 9.09 | 7.3 | 212 | 1.69 | 58.3 | 23.9 | 26.6 | 724.7 | 712.3 | 145.75 |
| JC15 | 8.78 | 7.3 | 54.2 | 2.98 | 111 | 29.4 | 92.5 | 146.4 | 382.3 | 277.5 |
| JC16 | 10.04 | 7.2 | 194 | 2.85 | 101 | 30.8 | 225.8 | 506.9 | 924.65 | 252.5 |
| JC17 | 8.99 | 7.3 | 40.1 | 2.64 | 101 | 23.3 | 25 | 377.6 | 390.85 | 252.5 |
| JC18 | 13.39 | 7.3 | 101 | 2.14 | 63.2 | 22.5 | 15.3 | 480.7 | 456.65 | 158 |
| JC19 | 8.81 | 7.5 | 71.7 | 3.33 | 123 | 31.4 | 143 | 368.4 | 589.33 | 307.5 |
| JC20 | 11.15 | 7 | 122 | 3.76 | 127 | 92.6 | 280.6 | 603.9 | 1003.94 | 317.5 |
| JC21 | 9.35 | 7.5 | 54.7 | 1.53 | 42.7 | 22.8 | 11.3 | 409.9 | 340.51 | 106.75 |
| JC22 | 8.89 | 7.5 | 56.7 | 1.71 | 55.1 | 28.5 | 7.1 | 379.4 | 346.55 | 137.75 |
| JC23 | 12.19 | 7.3 | 28.9 | 2.42 | 74.7 | 46.1 | 27.2 | 400.8 | 393.98 | 186.75 |
| JC24 | 9.47 | 7.1 | 42.7 | 2.9 | 108 | 32.3 | 67.5 | 363.6 | 450.89 | 270 |

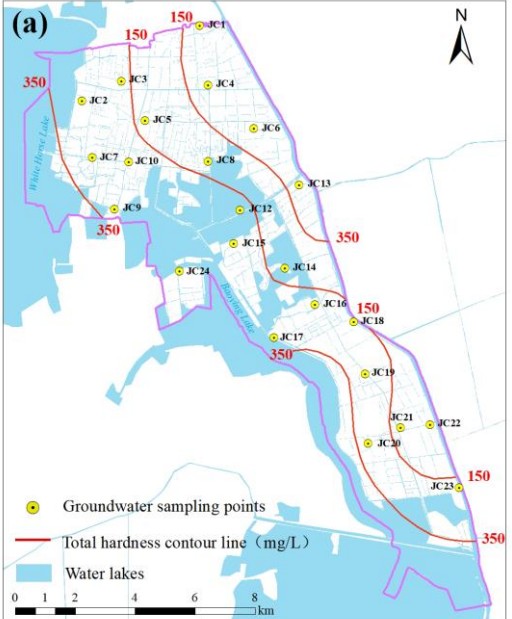 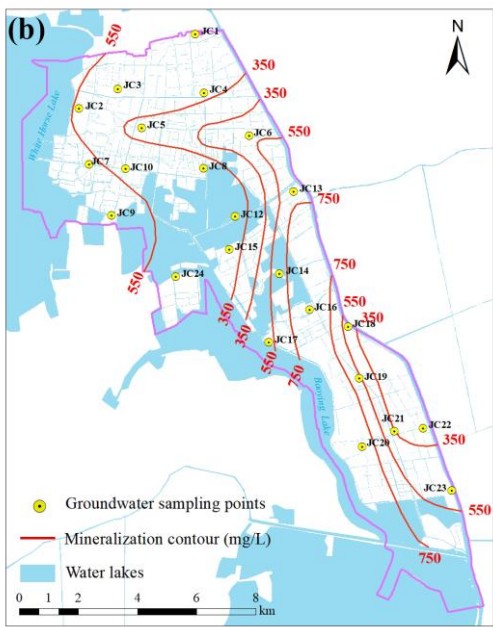

**Figure 7.** Groundwater quality contours for total hardness and mineralization. (**a**) Total hardness. (**b**) Mineralization.

## 4. Discussion

1.  Uneven intra-annual distribution of rainfall and high interannual variability

Water resources in the pilot evaluation area are largely formed by atmospheric rainfall. Rainfall varies considerably from year to year, with an annual maximum value of 1420.30 mm (1991) and a minimum value of 496.80 mm (1978). The ratio of maximum to minimum rainfall is 2.87. Within a year, rainfall is unevenly distributed and mostly concentrated during the flood season, with May to September accounting for 67.17% of the annual rainfall.

2.  Uneven intra-annual distribution of evaporation and decreasing total year by year

The intra-annual distribution of evaporation from the water surface is highly uneven due to temperature changes, i.e., high in summer and low in winter. Moreover, evaporation in the northern Yangtze River Delta has exhibited a decreasing trend over the years, decreasing by an average of 3.3 mm per year. On the one hand, the greenhouse effect caused by a large amount of industrialized pollution in the region over the past years has led to global warming, and temperature rise will, to a certain extent, lead to an increase in evaporation. On the other hand, the interaction between atmospheric pollution and specific climatic conditions creates a huge amount of hazy weather, which considerably reduces the intensity of insolation, and consequently, water surface evaporation. Therefore, the intensity of insolation exerts a greater influence on evaporation trend.

3.  Uneven temporal distribution and low availability of local runoff with high levels of transit water

Consistent with rainfall, local runoff is highly variable from year to year. The incoming water from the upper and middle reaches of the basin is largely concentrated during the flood season and mostly discharges into the river and sea, forming abandoned water without being utilized. In addition, transit water is an important part of local water resource utilization, primarily from the upper reaches of Yangtze River and Huai River rainfall and flood water disposal. The abundance of water in transit has been the greatest resource advantage of the region, providing unique conditions for its development.

4.  Overall good surface water and groundwater quality

The water quality of the Baoying section of the Grand Canal is relatively stable, i.e., II to III, and mostly II. The water quality of Baoying Lake and Baima Lake is III to IV, and the water quality of Yunxi River is III toIV. Most substandard months are during the non-flood season, and all of their values exceed the standard for permanganate index. Groundwater in the pilot evaluation area is Class III or better, with mineralization monitoring values ranging from 208.67 mg/L to 924.65 mg/L, and an average value of 517.81 mg/L. Therefore, this water is classified as freshwater, and water chemistry type is predominantly $HCO_3$-Ca·Na and $HCO_3$-Ca·Mg.

## 5. Conclusions

The northern Yangtze River Delta experiences significant variations in rainfall over time, in accordance with the analysis of multiyear rainfall and evaporation monitoring data. About 50% of the annual rainfall falls between June and August, causing frequent flooding. Floods and droughts still pose a hazard to the area, although water conservation projects have helped avert and alleviate disasters to a certain extent. The calculation of total water resources in the area shows that low local runoff and high transit water are the basic characteristics of water resources in the pilot area. Local water resources cannot meet water requirements, and although transit water is high, it varies considerably from year to year and cannot guarantee water security. Moreover, some problems with water wastage persist in the area. Meanwhile, the demand for water resources is constantly increasing with the rapid development of the local economy and society. Therefore, to guarantee sustainable development, drawing a red line for the development and use of water resources and strictly enforcing control on total water consumption are necessary.

In accordance with the water quality analysis of groundwater, the overall water quality assessment results are relatively good, indicating that water pollution control measures have been implemented well in recent years under the national "14th Five-Year Plan." However, as human living conditions continue to improve, the demands for water resources will increase in quantity and quality, and thus, laxity should have no room in water resource management. A comprehensive evaluation of water resources is a prerequisite for securing regional water resource management. In this regard, this study systematically presents the entire evaluation process by using the pilot area as an example. Future research will extend to the entire Yangtze River Delta, laying a solid foundation for the effective management

of water resources and promoting the sustainable and healthy development of Yangtze River Delta.

**Author Contributions:** Conceptualization, L.H.; Methodology, L.H.; Software, L.C.; Validation, C.X.; Formal analysis, L.C.; Data curation, S.L.; Writing—original draft, C.X.; Writing—review & editing, L.H. and S.C.; Visualization, C.X.; Supervision, S.C.; Project administration, S.L.; Funding acquisition, L.H., S.L. and S.C. All authors have read and agreed to the published version of the manuscript.

**Funding:** We are grateful to the funding support from the National Science Foundation of China (NFSC) (Nos. 42101384 and 41571386), the Nanjing Xiaozhuang University Nature Science High-level Research Project (Nos. 2022NXY03), the Natural Science Foundation of Jiangsu Province (BK20210043), the Water Conservancy Science and Technology Project of Jiangxi Province (202124ZDKT29), and the Research Foundation of Nanjing Hydraulic Research Institute (Y922003).

**Data Availability Statement:** Not applicable.

**Conflicts of Interest:** The authors declare no conflict of interest.

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
