# Peer review of "Comprehensive Evaluation of Water Resource Characteristics in the Northern Yangtze River Delta, China"

_water, doi:10.3390/w15061028_

Round 1
Reviewer 1 Report
This is an informative and well written manuscript reporting on the comprehensive evaluation of water resource characteristics. The research progress of this paper is novel and the research method is effective, which has important theoretical and practical significance, and provides important reference for follow-up research. Therefore, I would recommend a minor revision.
This paper is innovative in exploring the relatively well-developed north wing of the northern Yangtze River Delta, China as a pilot area and constructing a complete set of water resource evaluation models. By field data collection and analysis of the spatiotemporal changes in precipitation and evaporation and their influencing factors, the availability of various water resources was evaluated. At the same time, the overall water quality in the basin was evaluated, verifying the applicability of the evaluation model.
Here are the general comments from the reviewer:
1. Clarification is needed on the description of statistical items in Table 1, regarding the distinction between water surface area and paddy fields area, and whether Gaobao Lake area is included in the research area.
2. What is the data preprocessing method adopted by the author to the 29 water samples collected in the field?
3. The design of Figures 5 and 6 needs to be optimized and the legend part is not clear.
4. It is suggested to add more related references to enrich the research content of the paper, as some of them are quite outdated and have limited citation value.
Reviewer 2 Report
Comprehensive evaluation and analysis of the water resources characteristics of the Yangtze River Delta would be beneficial for promoting the sustainable utilization of water resources and realizing the coordinated development of basin economy, society, resources, and environment. The author's analysis of the water resources characteristics of the region is quite comprehensive. Research on both the quantity and quality of water resources was conducted objectively and comprehensively, with certain application value and research significance. The paper is well organized and has a clear logical structure. The remarkable part is that all the water quality samples were collected on site, which requires a large amount of funds and manpower, especially in the sampling of groundwater quality, making the data itself authentic and of certain value. However, it needs to be improved and supplemented with more details. Therefore, I provide suggestions for minor revision:
1. The references in the abstract part of the paper are relatively old, and it is necessary to study the current situation of the underground water resources evaluation model comprehensively and perfectly.
2. What is the basis for the selection of survey points when conducting water quality surveys in the selected study area?
3. The author also needs to consider applying other technologies, such as remote sensing technology, geological survey technology, and hydrological models, to assess the accessibility and usability of water resources more accurately.
4. In addition, to better assess the characteristics of water resources, the economic value of water resources and their social and environmental impacts should be considered. Meanwhile, water resources management and water resources protection policies should be considered seriously to evaluate water resources better comprehensively.
